# Variant Analysis and Strategic Clustering to Sub-Lineage of Double Mutant Strain B.1.617 of SARS-CoV-2

Vishal Mevada [1,*], Rajesh Patel [2,*], Pravin Dudhagara [2], Himani Gandhi [2], Urvisha Beladiya [2], Nilam Vaghamshi [2], Manoj Godhaniya [2] and Anjana Ghelani [3]

1    DNA Division, Directorate of Forensic Science, Gandhinagar 382007, India
2    Bioinformatics Laboratory, Super Computing Facility, Department of Biosciences, Veer Narmad South Gujarat University, Surat 395007, India; dudhagarapr@gmail.com (P.D.); himanigandhi8290@gmail.com (H.G.); urvishabeladiya3@gmail.com (U.B.); ndvaghmashi@vnsgu.ac.in (N.V.); mdgodhaniya@vnsgu.ac.in (M.G.)
3    Ramkrishna Institute of Computer Science and Applied Sciences, Surat 395007, India; ghelanianjana@gmail.com
*    Correspondence: vmevada102@gmail.com (V.M.); rkpatel@vnsgu.ac.in (R.P.); Tel.: +91-932-775-5074 (V.M.); +91-999-898-1741 (R.P.)

**Abstract:** SARS-CoV-2 is an RNA coronavirus responsible for Acute Respiratory Syndrome (COVID-19). In January 2021, the re-occurrence of COVID-19 infection was at its peak, considered the second wave of epidemics. In the initial stage, it was considered a double mutant strain due to two significant mutations observed in their Spike protein (E484Q and L452R). Although it was first detected in India later on, it was spread to several countries worldwide, causing high fatality due to this strain. In the present study, we investigated the spreading of B.1.617 strain worldwide through 822 genome sequences submitted in GISAID on 21 April 2021. All genome sequences were analyzed for variations in genome sequences based on their effects due to changes in nucleotides. At Allele frequency 0.05, there were a total of 47 variations in ORF1ab, 22 in Spike protein gene, 6 variations in N gene, 5 in ORF8 and M gene, four mutations in Orf7a, and one nucleotide substitution observed for ORF3a, ORF6 and ORF7b gene. The clustering for similar mutations mentioned B.1.617 sub-lineages. The outcome of this study established relative occurrence and spread worldwide. The study's finding represented that "double mutant" strain is not only spread through traveling but it is also observed to evolve naturally with different mutations observed in B.1.617 lineage. The information extracted from the study helps to understand viral evolution and genome variations of B.1.617 lineage. The results support the need of separating B.1.617 into sub-lineages.

**Keywords:** variant analysis; B.1.617; double mutant strain of SARS-CoV-2

## 1. Introduction

In late 2019, several people in Wuhan, China were infected with severe pneumonia at the hospitals. In a very short timespan, the infection was spread rapidly worldwide and designated as epidemics. The causative agent novel coronavirus, formerly known as "Wuhan seafood market pneumonia virus," first appeared at the seafood and wildlife wholesale market in Wuhan, Hubei Provence, China during late November/early December, 2019 [1]. As of 20 May 2021, a total of 167.85 million cases resulting in 3.49 million deaths in 215 countries have been confirmed [2]. While there are many thousands of variants of SARS-CoV-2, subtypes of the virus can be put into larger groupings such as lineages or clades [3]. Three main nomenclatures have been proposed, GISAID, Nextstrain and Phylogenetic Assignment of Named Global Outbreak Lineages (PANGOLIN) [4–6]. Recently, there were several cases reported rapidly in India during January 2021. The detailed study reported that the prevalence of lineage B.1.617, one of the known variants of SARS-CoV-2, rapidly increased from January to April 2021

in India [7]. It was first identified in Maharashtra, India, on 5 October 2020 [8]. It has also been referred to as a double mutation variant [9]. Although lineage B.1.617 was reported in India, more than 20 countries also reported several cases of the same lineage worldwide as on 21 April 2021. Initially, the WHO designated the strain as "variant of interest". However, after detailed assessment, it was assigned as a "variant of concern" to a lineage B.1.617 [10].

The genome of SARS-CoV-2 has a linear, positive-sense, single-stranded RNA genome about 30,000 bases long [11]. The genome has the highest composition of U (32.2%), followed by A (29.9%), and a similar composition of G (19.6%) and C (18.3%) [12]. Virus infections start when viral particles bind to host surface cellular receptors [13]. Several studies reported that two major mutations observed in this strain were E484Q and L452R. Both these mutations are concerning because they are located in a key portion of the viral spike protein region associated with penetration to human cells. The mortality from COVID-19 is higher in people older than 65 years and in people with underlying comorbidities, such as chronic lung disease, severe heart conditions, high blood pressure, obesity and diabetes [14]. Community transmission of the virus and random antiviral treatments allow higher mutation rates resulting in potentially virulent strain [15]. Therefore, systematic tracking of demographic distribution for such mutant strain is urgently required to combat COVID-19 infection effectively. In the natural environment, the mutation rate in RNA viruses is dramatically high, up to a million times higher than that of their hosts. This high rate correlates with virulence modulation and evolves adaptations in adverse conditions [16]. Wang and colleagues have characterized 13 variation sites in SARS-CoV-2 such as ORF1ab, S, ORF3a, ORF8 and N regions, positions 28,144 in ORF8 and 8782 in ORF1a showed a mutation rate of 30.53% and 29.47%, respectively [17]. Similarly, Maria Pachetti and co-workers identified mutations in ORF1ab (nsp2, nsp3, RdRp and nsp143), Spike protein gene and ORF9a (nucleocapsid and phosphoprotein) gene throughout their study [18]. The mutations observed in the genome may directly correlate with the efficacy of drugs in severe infections [19].

The present study analyzed 822 SARS-CoV-2 genome sequences from 20 countries belonging to Lineage B.1.617 for variations. This might be helpful to identify the emergence of subsequent clades from B.1.617 Lineage. The CovSurver, as well as SnpEff-based variant analysis, was performed for all the genome sequences submitted in GISAID database up to 21 April 2021.

## 2. Materials and Methods

### 2.1. Source and Selection of Samples

A total of 822 genome sequences of double mutant strain (B.1.617) in fasta format were downloaded on 21 April 2021, from Global Initiative on Sharing All Influenza Data (https://www.gisaid.org, accessed on 21 April 2021) [6]. A metadata file was downloaded from the same portal for mapping each sample history to the respective genome sequence. The reference genome Wuhan sequence (Accession No: NC_045512.2) was downloaded from the National Centre for Biotechnology Information (NCBI). Samples were selected based on pangolin lineage B.1.617, known as double mutant strain observed worldwide. All fasta sequences were mapped to metadata for Country-based grouping using the CLC Genomics Workbench 8.0 v tool (https://digitalinsights.qiagen.com, accessed on 15 May 2021). The quality of all sequences is approved through the FastQC tool (https://sourceforge.net/projects/fastqc.mirror, accessed on 10 May 2021).

### 2.2. Database Development for Mutations

Drupal 7.78-Based AnCOVID19 database was developed to analyse amino acid variations observed in the GISAID database. The metadata downloaded from https://www.epicov.org database (GISAID database) on 21 April 2021 [6] were extracted and uploaded to AnCovid19 Database for analysis. All 822 samples were further calculated for the frequency of each mutation using the Views 3.0 function of Drupal 7 (http://covid19.vnsguhpc.co.in, accessed on 25 May 2021).

### 2.3. Mapping against Reference Genome

All sequences were aligned using the Multiple sequence analysis tool Clustal Omega (https://www.ebi.ac.uk/Tools/msa/clustalo, accessed on 10 May 2021) [20] with Paramshavak Super Computing facility available at Department of Bioscience, Veer Narmad South Gujarat University. The resulted multifasta file was grouped based on the country for easy comparison in downstream analysis using CLC Genomics Workbench. The multifasta file was mapped against Severe Acute Respiratory Syndrome coronavirus two isolate Wuhan-Hu-1 genome (Accession No NC_045512.2) sequence using Bowtie sequence alignment tool [21]. The resulting BAM files were re-aligned to find the insert-Deletion from all genome sequences using the Insert indel quality tool available on http://usegalaxy.eu server [22], accessed on 15 June 2021.

### 2.4. Variant Identification Annotation

Variant calling was performed using the command line LoFreq call tool [23]. The resulting Variant Calling files were further processed via SnpEff [24] against the reference genome of Wuhan reference genome (NC_045512.2) downloaded from NCBI. The SnpEff tool was used from Galaxy web-based tool (http://usegalaxy.eu, accessed on 15 June 2021). The custom parameters include the setup of 5000 bases for Upstream/Downstream length and 2 bases of set size for splice sites (donors and acceptor) in bases were set in the SnpEff tool. The rest of the parameters were used as default values suggested by developers. SnpEff generates annotated VCF file along with the HTML report for each sample analyzed.

### 2.5. Comparative Variant Analysis

The resulting VCF files were uploaded to http://usegalaxy.eu (accessed on 15 June 2021) for comparative variant analysis. The "SnpSift Extract Fields" tool was used to extract "CHROM, POS, REF, ALT, DP, AF, DP4, SB, EFF(*).IMPACT EFF(*), FUNCLASS EFF(*), EFFECT EFF(*), GENE, CODON, Amino Acid Substitution", various effects from variant calling file [25]. The resulting tabular files were further analyzed for comparative statistical analysis using Linux-based GNU datamash tool (v.1.3) (https://www.gnu.org/software/datamash, accessed on 15 June 2021) [26]. The statistical analysis results were merged to a combined variation report using workflow explained in COVID-19: variation analysis reporting.

### 2.6. Data Visualization

The data were visualized using the Variant Frequency Plot tool to generate phylogenetic relations and plot for variants observed for each gene [27]. The dplyr v0.8.4 package (https://dplyr.tidyverse.org, accessed on 15 June 2021) was used to summarise the data for plotting [28]. The graphical representation was mentioned in Figure 1 for the complete workflow of the method.

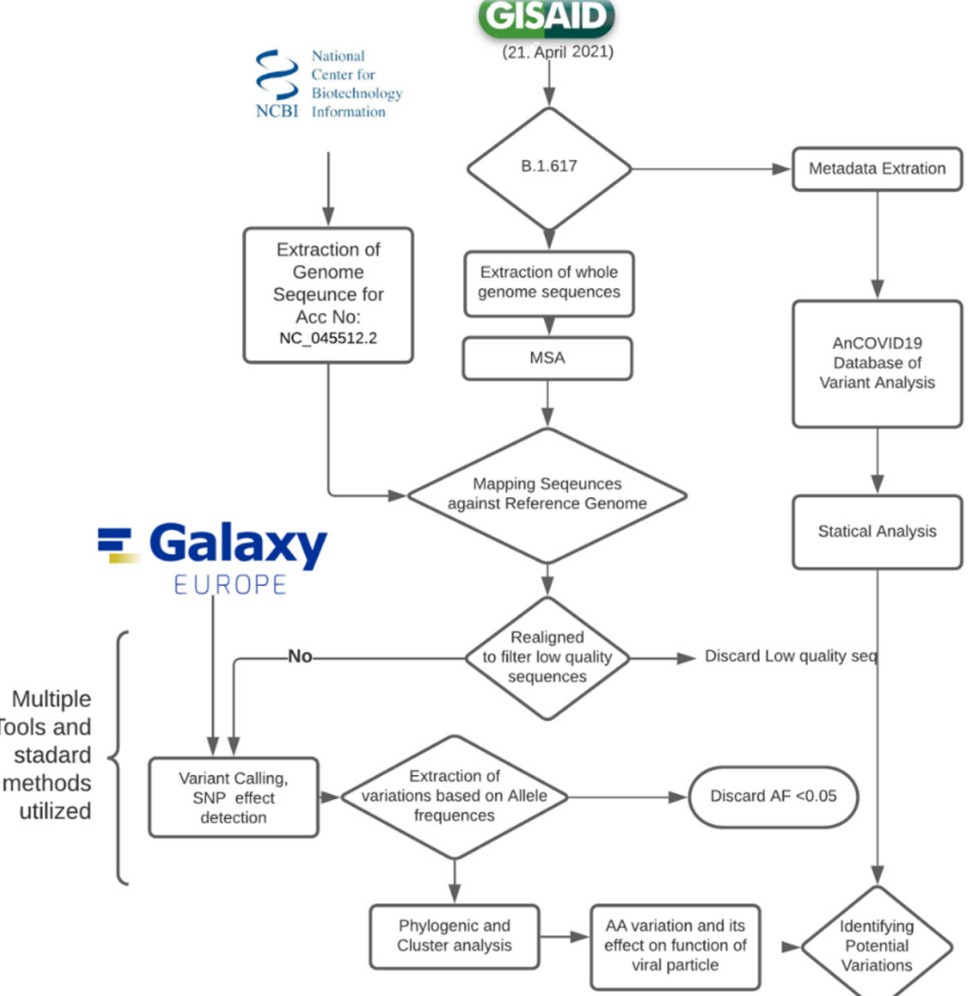

**Figure 1.** Flow chart for variant analysis protocol.

## 3. Results

Lineage B.1.617, also known as VUI (Variant Under Investigation), 2 April 2021, is one known variant of SARS-CoV-2, which causes COVID-19 [29]. It was first identified in India on 5 October 2020, and is a "double mutation" variant. "Double mutation" refers to B.1.617′s mutations in the SARS-CoV-2 spike protein's coding sequence at E484Q and L452R. This strain has been found in almost 20 different countries. The WHO announced that the world grapples to contain the surge in the COVID-19 cases, with 5.7 million infections detected in the third week of April 2021. The majority of patients were initially reported in Maharashtra, one of the fastest-growing states of India.

Our study shows that lineage B.1.617 was not unique to India for the first time and circulated in other countries from the USA, Singapore, England, Australia, Bahrain, etc. The lineage B.1.617 exhibits a medium prevalence in Russia, Arabia, and South African countries to date (Figure 2). The Genome sequence provides vital information for tracking and tracing infection worldwide.

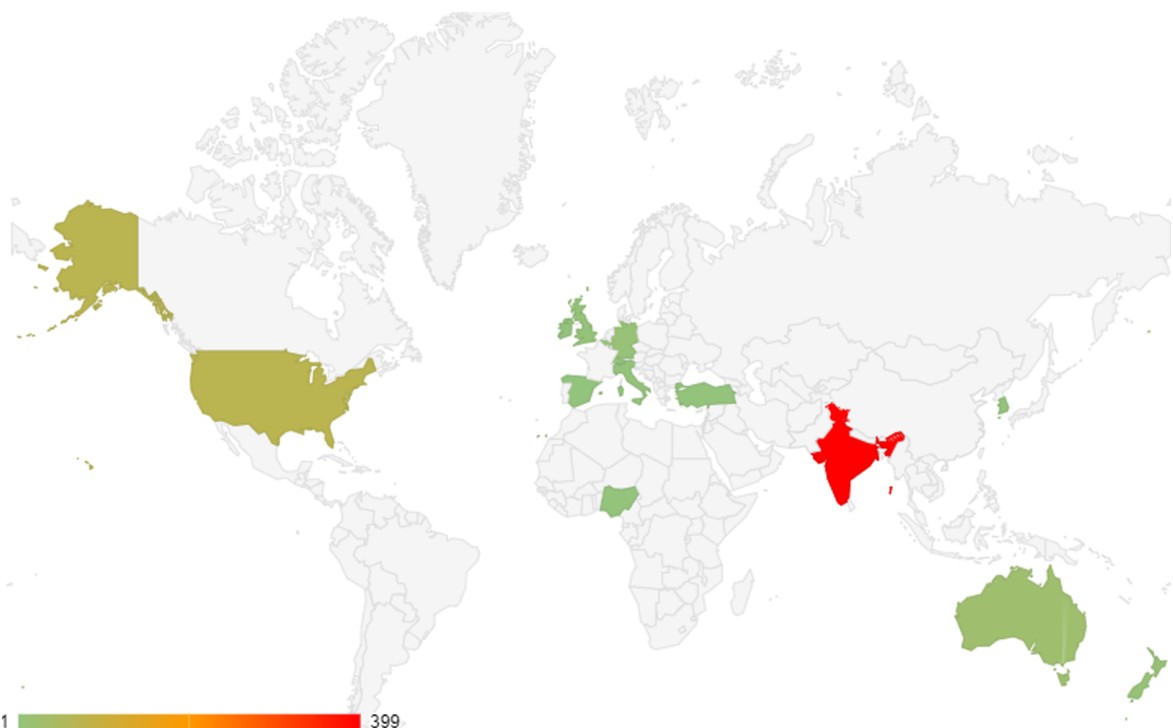

**Figure 2.** Worldwide distribution of B.1.617 strain of SARS-CoV-2 (21 April 2021).

Our analysis of over 822 genome sequences belonging to B.1.617 Lineage is available on the GISAID initiative up to the submission date of 21 April 2021. GISAID is a global science initiative and primary source established in 2008 that provides open access to genomic data of influenza viruses and the coronavirus responsible for the COVID-19 pandemic [6,30]. The results of FastQC indicated that all the genome sequences were not exactly similar to the original Wuhan stain (29,903 bp) used in the study. All the sequences were initially aligned using Clustal Omega to overcome this problem. The gaps were added with "-" symbol in each sequence. The resulting multifasta were separated based on the 20 countries. The number of sequences varies based on submission as of 21 April 2021. The maximum sequences for B.1.617 Lineage were submitted from India (399), followed by England (188), USA (72) etc. (Figure 2). All these sequences were further processed for SNP and Variant analysis.

### 3.1. Diversity of Mutations Observed in CovSurver (GISAID)

Along with Fasta sequences, the metadata file was downloaded for a detailed study of the samples submitted to GISAID. The metadata file was downloaded from CovSurver database. The metadata file was analyzed on AnCOVID19 database using a tool created using Views 3 of Drupal 7.79 (http://covid19.vnsguhpc.co.in:808, accessed on 21 April 2021). All these mutations were grouped into 24 different protein classes, as mentioned in the GISAID database. The analysis revealed 1772 other mutations observed in all genome sequences used in the present study. Out of 1772, the maximum mutations observed in Spike protein was 372, followed by NSP3, known as papain-like proteinase (234), Nucleocapsid (N) protein (198) and NSP4-nonstructural protein 4 (163). However, there were fewer mutations observed in NSP7 (11), NSP8 (12), NSP9 (7), NSP10 (12) and NSP12 (62) [31]. All these proteins are responsible for the replication and transcription of the viral genome. NS7a (ORF7a) and NS7b (ORF7b), known as highly conserved protein domains of the SARS-CoV-2 Genome, had 24 and 4 mutations, respectively [32,33]. One of the vital Membrane (M) Proteins observed 63 diverse mutations. In contrast, the Envelope (E) protein was observed with only four mutations amongst the whole dataset used in the present (Table 1). These mutations were observed based on sequence similarity in GISAID and Nextstrain-ncov databases (https://nextstrain.org/sars-cov-2,

accessed on 21 April 2021). A detailed account of these mutations is reported in the Supplementary Material Table S1. However, several studies reported silent mutation and a missense mutation in the SARS-CoV-2 genome sequence [34,35], not affecting the actual functionality of viral particles [34]. The genome sequence annotation and SNP detection were performed to understand the significant type of mutations observed throughout whole-genome sequences.

*3.2. Mapping of Sequences to Reference Genome*

The multifasta files generated from ClustalO were separated and mapped against the Wuhan-Hu-1 reference genome (Accession No: NC_045512.2) using the Bowite2 tool. The resulting BAM files were grouped based on the country and uploaded to the http://usegalaxy.eu (accessed on 21 April 2021) portal for further analysis using modified workflow published on Usegalaxy portal [36]. This is the first instance reported to directly process whole-genome sequence for variant analysis using this workflow. One can download the modified workflow from the AnCOVID19 web portal (http://covid19.vnsguhpc.co.in:808, accessed on 30 June 2021). The resulting files were re-aligned to remove duplicate sequences from the inputs using Re-align reads on the Galaxy server. All the samples were first processed for insertion-deletion type mutations by adding indel qualities with lofreq Insert indel qualities tool followed by using the lofreq Call variants tool. The lofreq insert indel qualities insert adequate quality score as reported in the re-aligned sequences. While the lofreq call variant tool identifies the variations based on the quality score and nucleotide alignment.

**Table 1.** Number of Sequence/s and mutations observed using CoVsurver (GISAID) Metadata.

| Sr No. | Country | No. of Genome Sequences | No. of Mutations Observed in CoVsurver (GISAID) Metadata Analyzed Using AnCOVID19 Database | | | | | | | | | | | | | | | | | | | | | | | |
| | | | E Protein | M Protein | NS3 | NS6 | NS7a | NS7b | NSP8 | NSP10 | NSP12 | NSP13 | NSP14 | NSP15 | NSP16 | NSP1 | NSP2 | NSP3 | NSP4 | NSP5 | NSP6 | NSP7 | NSP8 | NSP9 | N | Spike |
|---|---|---|---|---|---|---|---|---|---|---|---|---|---|---|---|---|---|---|---|---|---|---|---|---|---|---|
| 1 | Australia | 23 | 0 | 2 | 4 | 2 | 3 | 1 | 6 | 0 | 6 | 5 | 6 | 9 | 3 | 2 | 3 | 10 | 7 | 1 | 4 | 0 | 0 | 0 | 6 | 25 |
| 2 | Bahrain | 22 | 1 | 3 | 6 | 1 | 4 | 1 | 3 | 0 | 7 | 3 | 3 | 4 | 1 | 0 | 4 | 11 | 3 | 1 | 4 | 1 | 0 | 1 | 10 | 22 |
| 3 | Belgium | 4 | 2 | 1 | 1 | 1 | 1 | 0 | 1 | 0 | 2 | 2 | 0 | 1 | 0 | 0 | 0 | 5 | 0 | 0 | 1 | 0 | 0 | 0 | 5 | 11 |
| 4 | England | 188 | 2 | 4 | 16 | 11 | 10 | 3 | 10 | 4 | 15 | 19 | 10 | 5 | 4 | 13 | 13 | 34 | 17 | 3 | 7 | 3 | 3 | 3 | 25 | 44 |
| 5 | Germany | 11 | 0 | 2 | 3 | 1 | 3 | 1 | 1 | 0 | 4 | 4 | 1 | 4 | 2 | 1 | 2 | 6 | 4 | 0 | 2 | 2 | 0 | 0 | 8 | 28 |
| 6 | Guadeloupe | 2 | 0 | 1 | 1 | 0 | 1 | 0 | 1 | 0 | 1 | 2 | 0 | 3 | 0 | 0 | 0 | 1 | 0 | 0 | 1 | 0 | 1 | 0 | 2 | 7 |
| 7 | India | 399 | 4 | 53 | 57 | 4 | 17 | 2 | 21 | 6 | 34 | 86 | 45 | 50 | 13 | 13 | 21 | 159 | 140 | 52 | 55 | 2 | 4 | 2 | 165 | 311 |
| 8 | Ireland | 3 | 0 | 1 | 2 | 1 | 1 | 0 | 2 | 0 | 1 | 2 | 1 | 2 | 0 | 0 | 0 | 2 | 0 | 0 | 1 | 0 | 0 | 0 | 3 | 9 |
| 9 | Italy | 3 | 1 | 1 | 2 | 0 | 2 | 1 | 0 | 0 | 3 | 2 | 1 | 0 | 0 | 0 | 2 | 8 | 3 | 0 | 1 | 0 | 0 | 0 | 6 | 14 |
| 10 | New Zealand | 11 | 1 | 2 | 2 | 0 | 4 | 1 | 0 | 0 | 4 | 5 | 2 | 4 | 1 | 0 | 3 | 7 | 3 | 0 | 3 | 0 | 0 | 0 | 6 | 15 |
| 11 | Nigeria | 3 | 1 | 1 | 1 | 0 | 0 | 0 | 0 | 0 | 1 | 0 | 0 | 0 | 0 | 0 | 1 | 2 | 0 | 0 | 0 | 0 | 1 | 0 | 2 | 7 |
| 12 | Scotland | 10 | 0 | 2 | 3 | 2 | 2 | 1 | 2 | 0 | 4 | 4 | 1 | 2 | 1 | 0 | 0 | 5 | 3 | 0 | 3 | 0 | 0 | 0 | 6 | 21 |
| 13 | Singapore | 50 | 0 | 5 | 6 | 13 | 6 | 1 | 7 | 0 | 7 | 6 | 10 | 6 | 3 | 2 | 11 | 13 | 6 | 3 | 5 | 1 | 2 | 0 | 7 | 29 |
| 14 | Sint Maarten | 1 | 0 | 0 | 1 | 1 | 2 | 0 | 0 | 0 | 2 | 2 | 0 | 2 | 0 | 1 | 0 | 1 | 0 | 0 | 2 | 0 | 0 | 0 | 2 | 9 |
| 15 | South Korea | 1 | 0 | 1 | 1 | 1 | 2 | 0 | 2 | 1 | 1 | 2 | 1 | 3 | 1 | 0 | 1 | 2 | 1 | 1 | 1 | 0 | 0 | 1 | 2 | 11 |
| 16 | Spain | 1 | 0 | 1 | 0 | 0 | 0 | 0 | 0 | 0 | 1 | 1 | 0 | 0 | 0 | 0 | 0 | 0 | 0 | 0 | 0 | 0 | 0 | 0 | 2 | 2 |
| 17 | Switzerland | 5 | 0 | 2 | 2 | 0 | 4 | 0 | 0 | 0 | 2 | 4 | 0 | 3 | 0 | 0 | 2 | 9 | 2 | 0 | 6 | 0 | 0 | 0 | 7 | 20 |
| 18 | Turkey | 2 | 0 | 2 | 1 | 0 | 0 | 0 | 1 | 0 | 2 | 0 | 0 | 0 | 0 | 0 | 1 | 4 | 1 | 0 | 0 | 0 | 0 | 0 | 3 | 4 |
| 19 | USA | 72 | 1 | 5 | 7 | 3 | 8 | 1 | 6 | 0 | 10 | 8 | 4 | 8 | 3 | 2 | 8 | 30 | 7 | 3 | 6 | 1 | 1 | 0 | 19 | 35 |
| 20 | Wales | 7 | 0 | 2 | 2 | 1 | 5 | 0 | 1 | 1 | 4 | 5 | 1 | 3 | 1 | 1 | 1 | 3 | 1 | 0 | 2 | 1 | 0 | 0 | 4 | 18 |

### 3.3. Variance Annotation

The Variant Calling Files (VCF) files were processed for the effect of SNP variation with SnpEff eff tool available in Galaxy Europe. The workflow generates a Variant Calling File (VCF) for reporting purposes having types of mutations, Allele frequency, and annotated SNP variations. Based on allele frequency and number of sequences, eight countries with fewer sequences were dropped from the further analysis. Whereas, the remaining sequences from 12 countries were processed for Variant Calling Reporting. Initially, the fields were extracted and converted to tabular format using SnpEff Extract tool integrated with usegalaxy portal.

In the present study, the highest variations were observed for England (122) followed by Singapore (85), India (78), New Zealand-Scotland-Bahrain (69), USA (63), Australia (57), Germany (50), Switzerland (48), Wales (42) and South Korea (33) (Table 2). The primary types of mutation include Single Nucleotide Polymorphism (SNP), Insertion (INS) and Deletion (DEL). SnpEff algorithm identified the effects out of the variations based on functional annotations.

**Table 2.** Variations and effects observed in each country.

| Country | Total Variations | Type of Mutation | | | Observed Variation Rate | Number of Effects |
|---|---|---|---|---|---|---|
| | | **SNP** | **INS** | **DEL** | | |
| Australia | 57 | 54 | 0 | 3 | 524 | 104 |
| Bahrain | 69 | 53 | 8 | 8 | 433 | 131 |
| England | 122 | 110 | 4 | 8 | 245 | 268 |
| Germany | 50 | 46 | 0 | 4 | 598 | 83 |
| India | 78 | 63 | 0 | 15 | 383 | 152 |
| New Zealand | 69 | 65 | 1 | 3 | 433 | 156 |
| Scotland | 69 | 64 | 1 | 4 | 433 | 126 |
| Singapore | 85 | 78 | 2 | 5 | 351 | 170 |
| South Korea | 33 | 32 | 0 | 1 | 906 | 52 |
| Switzerland | 48 | 44 | 1 | 3 | 622 | 96 |
| USA | 63 | 42 | 0 | 21 | 474 | 143 |
| Wales | 42 | 40 | 0 | 2 | 711 | 73 |

(SNP: Single Nucleotide Polymorphism, INS: Insertion, DEL: Deletion).

The annotations amongst the samples were grouped into four major classes (Modifier, Moderate, Low, High impact) based on the impact of mutations. The sequences from all countries (54–69%) were found to have a moderate type of impact (Figure 3). With respect to that, the low impact assisted mutations range from 12.59% to 41.67%. Both these types of impact might show moderate to low effect on cell function at the molecular level. However, the modifier type of effect (2.24−7.69%) and High impact effects (0.59−27.27%) show major concern regarding the function of the virus particle. The high impact effects were mainly reported in USA (27.27%), Bahrain (10.69%), England (7.46%), Scotland (7.14%), India (6.58%) and Singapore (0.59%). Based on the functional classification of mutations, Missense mutations in prime level (55.06−76.92%) were followed by silent mutations (23.08–44.94%). Meanwhile, only the sequences of Scotland and Singapore were found to have nonsense mutations 0.90% and 0.63%, respectively (Figure 4).

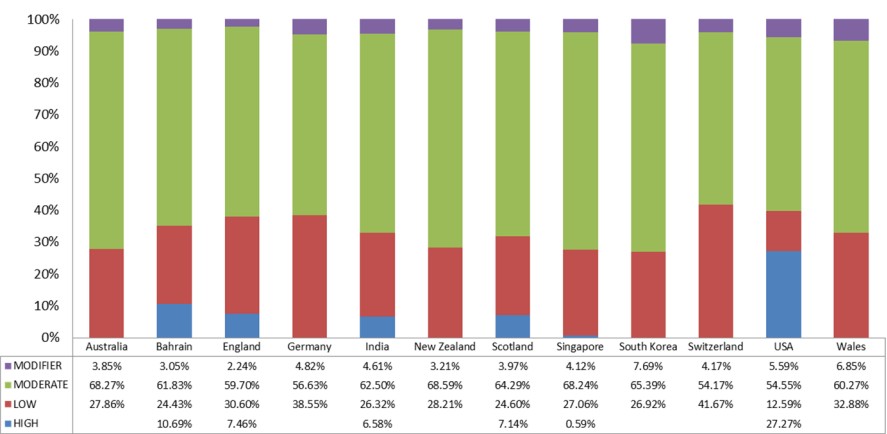

| | Australia | Bahrain | England | Germany | India | New Zealand | Scotland | Singapore | South Korea | Switzerland | USA | Wales |
|---|---|---|---|---|---|---|---|---|---|---|---|---|
| MODIFIER | 3.85% | 3.05% | 2.24% | 4.82% | 4.61% | 3.21% | 3.97% | 4.12% | 7.69% | 4.17% | 5.59% | 6.85% |
| MODERATE | 68.27% | 61.83% | 59.70% | 56.63% | 62.50% | 68.59% | 64.29% | 68.24% | 65.39% | 54.17% | 54.55% | 60.27% |
| LOW | 27.86% | 24.43% | 30.60% | 38.55% | 26.32% | 28.21% | 24.60% | 27.06% | 26.92% | 41.67% | 12.59% | 32.88% |
| HIGH | | 10.69% | 7.46% | | 6.58% | | 7.14% | 0.59% | | | 27.27% | |

**Figure 3.** Effects of mutations grouped by their impact.

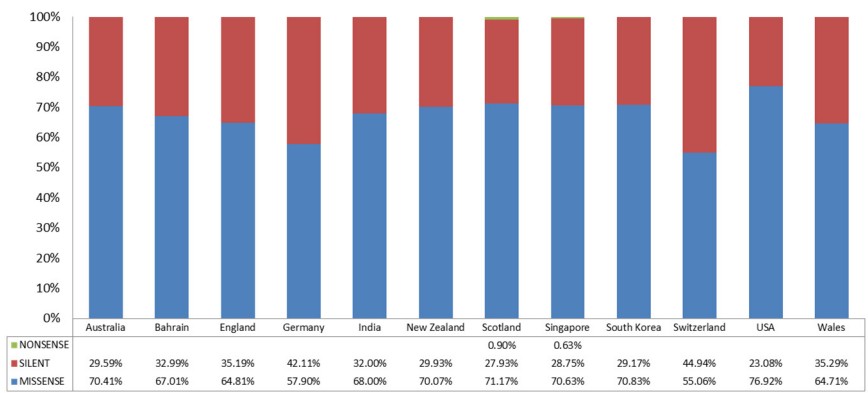

| | Australia | Bahrain | England | Germany | India | New Zealand | Scotland | Singapore | South Korea | Switzerland | USA | Wales |
|---|---|---|---|---|---|---|---|---|---|---|---|---|
| NONSENSE | | | | | | | 0.90% | 0.63% | | | | |
| SILENT | 29.59% | 32.99% | 35.19% | 42.11% | 32.00% | 29.93% | 27.93% | 28.75% | 29.17% | 44.94% | 23.08% | 35.29% |
| MISSENSE | 70.41% | 67.01% | 64.81% | 57.90% | 68.00% | 70.07% | 71.17% | 70.63% | 70.83% | 55.06% | 76.92% | 64.71% |

**Figure 4.** Effects of mutations grouped by type of mutation.

The variations reported were further grouped to type effects and regions where the variation was observed in genome sequences. A total of 11 categories were reported in processed sequences. Most of the mutations were observed as non-synonymous in the coding region ranging from 40.82% to 66.35% (Table 3). The second category of effects to the region includes Synonymous coding (12.25% in USA to 41.67% in Switzerland) followed by Frameshift (6.35% in Scotland to 26.53% in USA) and intergenic type (2.21% in England to 6.85% in Wales). At the same time, codon changes with codon insertion-deletion and codon deletion were reported with nearly similar amounts. However, codon Insertion, Start Lost and mutation in Splice site region reported in only one country, Bahrain, USA and Singapore, respectively.

**Table 3.** Comparative ratio for type of effects and region observed for each countries.

| No. | Type of Effect and Region | Australia | Bahrain | England | Germany | India | New Zealand | Scotland | Singapore | South Korea | Switzerland | USA | Wales |
|-----|---------------------------|-----------|---------|---------|---------|-------|-------------|----------|-----------|-------------|-------------|-----|-------|
| 1 | Codon change and Codon deletion | 0.96% | 3.82% | 0.74% | 3.61% | 5.92% | 0.64% | 0.79% | 1.16% | - | 1.04% | 8.84% | - |
| 2 | Codon change and Codon insertion | - | 6.87% | 0.37% | - | - | 1.28% | - | - | - | 1.04% | - | - |
| 3 | Codon deletion | 0.96% | 0.76% | 2.21% | - | 0.66% | 0.64% | 0.79% | 0.58% | - | 1.04% | 3.40% | - |
| 4 | Codon insertion | - | 0.76% | - | - | - | - | - | - | - | - | - | - |
| 5 | Frame shift | - | 10.69% | 7.35% | - | 6.58% | - | 6.35% | - | - | - | 26.53% | - |
| 6 | Intergenic | 3.85% | 3.05% | 2.21% | 4.82% | 4.61% | 3.21% | 3.97% | 4.07% | 7.69% | 4.17% | 5.44% | 6.85% |
| 7 | Non synonymous coding | 66.35% | 49.62% | 55.52% | 53.01% | 55.92% | 66.03% | 62.70% | 65.70% | 65.39% | 51.04% | 40.82% | 60.27% |
| 8 | Splice site region | - | - | - | - | - | - | - | 1.16% | - | - | - | - |
| 9 | Start lost | - | - | - | - | - | - | - | - | - | - | 2.72% | - |
| 10 | Stop gained | - | - | 1.47% | - | - | - | 0.79% | 0.58% | - | - | - | - |
| 11 | Synonymous coding | 27.89% | 24.43% | 30.15% | 38.55% | 26.32% | 28.21% | 24.60% | 26.74% | 26.92% | 41.67% | 12.25% | 32.88% |

### 3.4. Statistical Analysis Based on Allele Frequency

The NextGen sequencing methods also show some errors in sequencing. This may lead to false-positive mutations. The allele frequency-based statistical analysis was applied to minimize such errors in the samples. With respect to that, all mutations were screened for four different allele frequencies (>0.5, >0.1, >0.05 and >0.01). The outcome represents a slight difference observed in the case of AF > 0.5 and AF > 0.1 in all regions. However, a significant difference was observed for allele frequency above 0.05 compared to 0.5 and 0.1. Meanwhile, the AF > 0.01 may include false positive variations. The Allele frequency comparison allows us to define a strategy for selecting the range of Allele frequency observed. In the present study, the allele frequency lower than 0.05 was skipped for further research—a total of 92 variations reported above allele frequency 0.05. Out of 92, maximum (47) variations were observed in the ORF1ab gene, followed by 22 mutations in the Spike protein gene. Although the higher mutations were reported in the ORF1ab gene, the major concern was found in the Spike protein gene (S) (Figure 5).

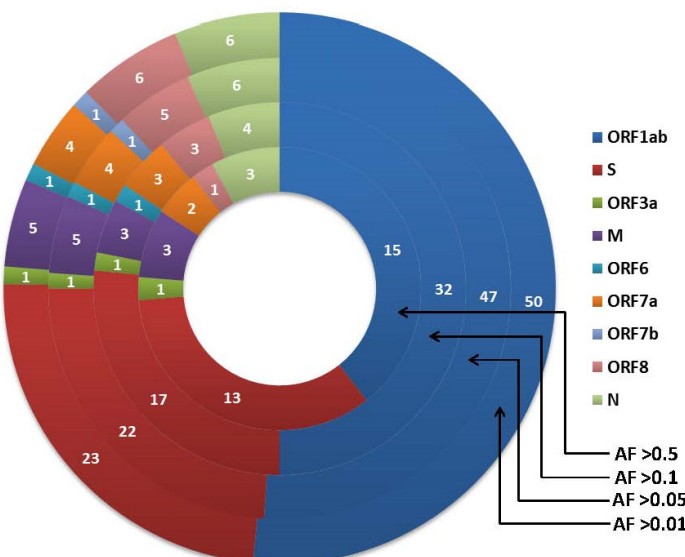

**Figure 5.** Comparative occurrence of variations observed for different allele frequency. (All the samples were analysed at Allele frequency (AF) 0.5, 0.1, 0.05, and 0.01. Higher Allele frequency corresponds to high degree of effects based on mutations in their genome. Mutations at lower AF (<0.01) are likely to observed due to false positive mutations. The major mutations observed in ORF1ab (15), followed by Spike Protein(13), ORF3a(3) and Membrane Glycoprotein (3) at allele frequency > 0.5).

Out of 47 variations observed in ORF1ab Gene, 14408-C > T (P4715L) mutation was reported for all 12 countries. While 11 countries reported variations in 3457-C > T (Y1064), 4965-C > T (T1567I), 11201-A > G (T3646A), 16852-G > T (G5530C), 17523-G > T (M5753I), 20396-A > G (K6711R) and 20401-T > G (S6713A). Amongst all variations observed in ORF1ab, 14408-C > T (P4715L), 14429-T > C (V4722A), 14874-G > T (K4870N), 15451-G > A (G5063S), and 15463-G > A (V5067I) codes for RNA-dependent RNA polymerase (RdRp). Whereas the changes observed 16375-C > T (P5371S), 16466-C > T (P5401L), 16852-G > T (G5530C), 17385-T > G (D5707E) and 17523-G > T (M5753I) were associated with a coding sequence for Helicase of SARS-CoV-2 genome (Figure 6).

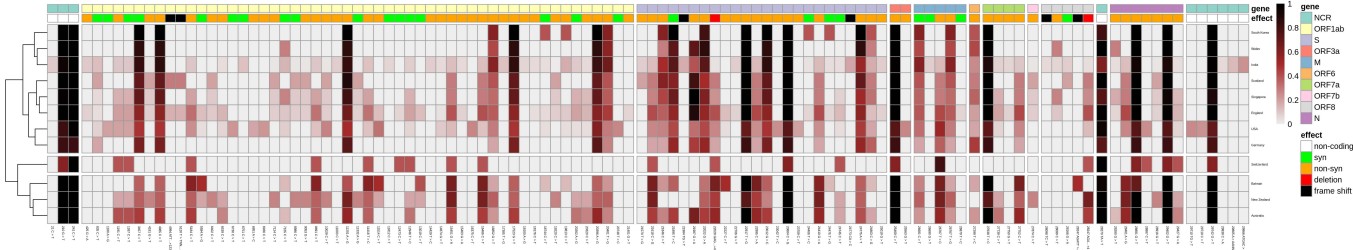

**Figure 6.** Phylogenetic comparison and Heatmap analysis of variations observed.

In the case of Spike protein, 23604-C > G (P681R) was reported in all 12 samples. Whereas 22022-G > A (E154K), 22917-T > G (L452R), 23012-G > C (E484Q) and 24775-A > T (Q1071H) reported in 11 samples. Out of these four mutations in the Spike protein 22917-T > G (L452R) and 23012-G > C (E484Q) were highly associated with the Human Ace 2 Protein and Spike Protein binding affinity [37]. Amongst all the samples, 10 countries reported 21618-C > G (T19R) and 21846-C > T (T95I) missense mutation. The missense mutations 22995-C > A (T478K), 24410-G > A (D950N) and 24863-C > G (H1101D) were also reported from samples from eight different countries.

Similarly, all 12 were countries found to have 28881-G > T (R203M) and 29402-G > T (D377Y) in N gene. The different missense type of mutation were observed on 26767 positions, 26767-T > C (I82T) and 26767-T > G (I82S) for 10 and 11 samples, respectively. In ORF8 only a single missense mutation was observed 28099-C > T (S69L) in England, Singapore and Wales. The variations 27638-T > C (V82A), 27739-C > T (L116F), 27750-G > T (K119N) and 27752-C > T (T120I) reported for ORF7a. ORF7a is a transmembrane protein with an N-terminal immunoglobulin-like ectodomain that consists of two β sheets held together by two disulfide bonds. The mutation in ORF7a causes destabilization of protein structure and enhances hindrance from Human Immune response [38]. The mutations in ORF7b at position 27874-C > T (T40I), ORF6 at position 27299-T > C (I33T) and ORF3a at 25469-C > T (S26L) were reported. Among them, mutations at ORF6 and ORF7b were considered rare mutations [39] (Supplementary File S1).

The most common base change observed was C<->T (335) transition type mutation, followed by G<->T (165) transversion type of mutation and A<->G (119) transition type of mutation (Figure 7). Amongst all base substitutions, a total of 454 transitions and 237 transversion types of base change were observed. The cumulative 1.915 ratios were observed for Transition/Transversion type of based pair changes. The amino acid substitution clearly defined the highest non-synonymous type of mutation reported as Threonine from Isoluecine (121) followed by Leucine > Proline (119), Valine > Alanine (46), Alanine > Theonine (46) and Alanine> Valine (41) amino acids (Figure 8).

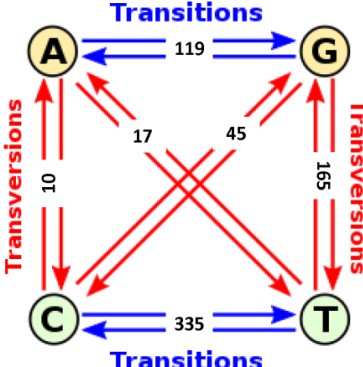

**Figure 7.** Base changes and type of mutations observed amongst samples. (Red colour arrow represents the Transversion type of mutations, Blue colour arrow indicates the Transition type of mutations. Total 237 transversion type mutations and 454 transition type of mutations observed in samples used in present study).

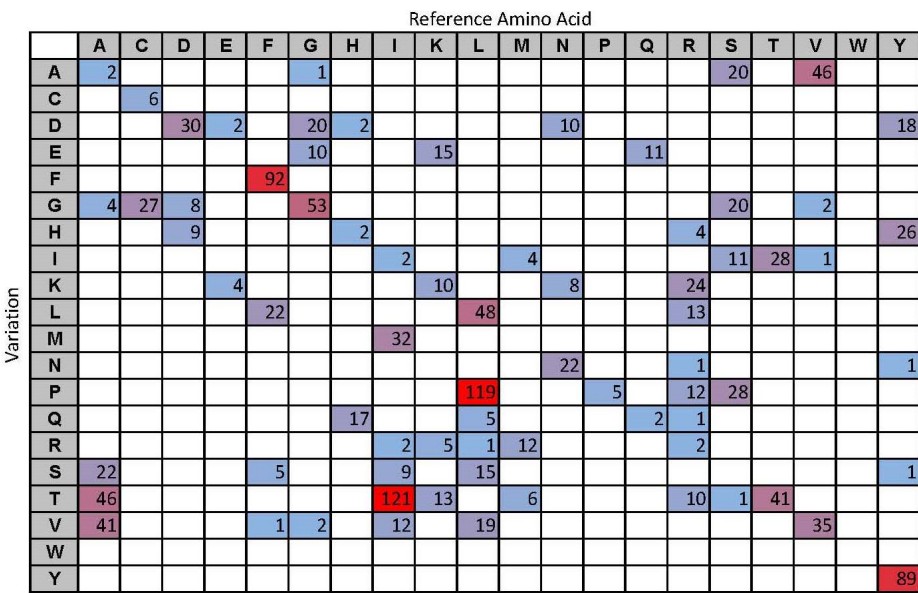

**Figure 8.** Amino acid substitution. (The gradient of colour High (red) to Low (blue) was applied to the number of variations to respective amino-acids).

### 3.5. Phylogenetic and Cluster Analysis

The combined analysis of all 12 countries represents the highest number of mutations in amino acids observed in the ORF1ab followed by spike proteins, Nucleocapsid Protein. All the results were grouped into clusters of 3, 4 and 5 based on mutation observed in their sequences (Figure 9). Upon clustering the variations in the cluster of three, similar mutations were observed for England, India, Germany, Singapore, South Korea, the USA and Wales. The pattern of mutations was highly identical to India for all the remaining seven countries. While the second group reported a total of three countries including Australia, Bahrain and New Zealand. However, the Nucleotide substitution pattern observed for the S gene in Switzerland was different from both groups. Similarly, the first two clusters for 4 and 5 remain similar to 3, placing Australia, Bahrain and New Zealand in the first and Singapore in the second cluster. Despite that, the third cluster of 3 was fragmented in 2 and 3 clusters upon grouping with 4 and 5 clusters, respectively.

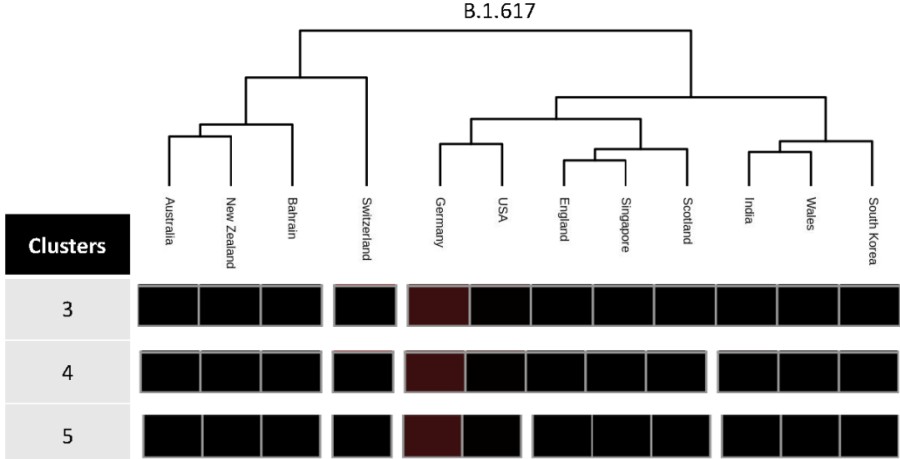

**Figure 9.** Comparison of variations based on clusters observed in similarity. (Based on similar mutations observed in each country, Phylogenic chart suggesting the subsequence classification of the double mutant strain to 3, 4, and 5 clusters. The similarity represented in black colored highly similar sequences followed by brown color).

## 4. Discussion

"Double mutation" refers to B.1.617's mutations in the gene encoding the SARS-CoV-2 spike protein causing the substitutions E484Q and L452R [40]. It is identified as the 21A clade under the Nextstrain phylogenetic classification system [37]. The research study suggests that the variant may be more transmissible than previously evolved [41]. The data received from the Indian government's Integrated Disease Surveillance Programme (IDSP) stated that 32% of patients were below the age of 30 in the second wave compared to 31% in the first wave. Amongst infected people, more than 54% of patients need external Oxygen supply compared to only 41% in first wave of infection. The infection caused by B.1.617 led to an increased fatality rate from January to April 2021. The strain was first detected during December 2021 in India. However, the strain was reported in geographic areas of 20 different countries as of 21 April 2021.

In the present work, we have compared the SARS-CoV-2 genomes of B.1.617 lineage to reference Wuhan's seafood market genome sequence with the aim of gaining important insights into virus mutations. Initially, the major concern of this strain was about two mutations observed (E484Q and L452R) in Spike protein region. Over time, mutations D614G, D950N, E154K, G142D, H1101D, P681R, Q1071H and T95I in Spike protein were showing higher ratio in several studies similar to our results [42,43]. All these mutations might allow the virus to easily integrate via human ACE2 receptor causing a drastic influence on the treatment with antiviral drugs [44]. Qianqian Li and coworkers also reported the mutations caused on the D614G site make the virus more infectious compared to wild type [45]. Along with substitution type of mutations, Zhe Liu observed deletion mutations that affect the polybasic cleavage site itself (NSPRRAR) or a flanking sequence (QTQTN) [46]. This mutation was beneficial to study the cross-species host transfer mechanism from SARS-CoV-1 to SARS-CoV-2.

The second highest mutations were reported for ORF1ab gene. ORF1ab codes for multiple proteins (RdRP, Helicases and PLPro), which guide the infection process after invading the cell. Aayatti Mallick Gupta et al. reported the ancestral viral samples from China and most parts of Asia, isolated since the initial outbreak and the later evolved variants isolated from Europe and the Americas in the case of P4715L non-synonymous mutation [47]. The P4715L mutation was reported in the highest number in our study. Begum et al. studied P4715L mutation directly affecting the molecular structure of RdRp [48]. All these mutations observed in ORF1ab region directly associated with reduced drug efficacy for currently administered drugs for RdRp [49]. Along with that, the mutations observed on location 16375-C > T (P5371S), 16466-C > T (P5401L), 16852-G > T (G5530C), 17385-T > G (D5707E) and 17523-G > T (M5753I) were associated with coding sequence for Helicase of SARS-CoV-2 genome. The computational analyses indicated the possible role of these mutations in enhancing the affinity of helicase RNA interaction and hence replication [50].

The N protein involved in the viral assembly, replication, and regulation of host immune response is a major structural component of SARS-CoV-2 and plays essential roles in the viral life cycle. These characteristics make the N protein a necessary target for viral diagnosis and vaccine development [51]. N gene Insertions are known to be a very rare type of mutation, which account for less than 0.1% of detected SARS-CoV-2 mutation cases. In contrast, in-frame deletions that reduce the length of the viral N protein without using stop codons account for about 0.6% of detected viral mutation cases [52]. The Nextstrain SARS-CoV-2 resources identified novel 12 nt deletions observed at positions located between 28890–28901 at the variable region of the viral N Gene. Concerning that, our results were beyond observed in Nextstrain resources at locations of 28881-G > T (R203M) and 29402-G > T (D377Y) for non-synonymous mutations. The detection of N gene is an important criterion for the diagnosis of the SARS-CoV-2 in various commercial kits. As per the recommendations from the US Food and Drug Administration department, the presence of SARS-CoV-2 genetic variants in a patient sample can potentially change the performance of the SARS-CoV-2 test. The prevalence of genetic variants in N Gene may lead to more false negative results than otherwise expected in that case.

ORF7a is a transmembrane protein with an N-terminal immunoglobulin-like ectodomain that consists of two β sheets held together by two disulfide bonds. The mutation in ORF7a causes destabilization of protein structure and enhances hindrance from Human Immune response [38]. The study on the African population identified ORF7a, ORF7b and ORF10 classified as the conserved gene [53]. However, some of the mutations for this double mutant strain were observed on sites 27638-T > C (V82A), 27739-C > T (L116F), 27750-G > T (K119N) and 27752-C > T (T120I). The previous study reported that the ORF8 protein inhibits the presentation of viral antigens by the major histocompatibility complex class I (MHC-I) and interacts with host factors involved in pulmonary inflammation. Amongst contributing mutations, Q27STOP, a mutation in the ORF8 protein defines the B.1.1.7 lineage of SARS-CoV-2, which engenders the second wave of COVID-19 [54]. While only single missense mutation was observed at region 28099-C > T (S69L) in sequences from England, Singapore and Wales for B.1.617.

So far, all these mutations have not been detected in more than 20 countries. The number and the occurrence, as well as the median value of virus point mutations registered out of Asia increase over time, propagating the increased cases of SARS-CoV-2. All these mutations were responsible for the second wave of this epidemic, causing an increased fatality rate in India and other countries. The major types of mutations observed for Non-synonymous coding regions may efficiently hinder and change interactions with human hosts.

On 4 May 2021, the CDC, in coordination with the SARS-CoV-2 Interagency Group (SIG), sub-divided B.1.617 to three lineages, B.1.617.1, B.1.617.2, and B.1.617.3. As per World Health Organization (WHO), it has said that only B.1.617.2 is now "Variant of Concern". The rest of two lineages were excluded from the list of a variant of concern. The B.1.617.2 is now labelled variant Delta and has been reported in 62 countries as of June 1, 2021. GISAID data showed that B.1.617.1 and B.1.617.2 together account for 70% of all SARS-CoV-2 genomes sampled in India. B.1.617.2 appears to be particularly gaining prevalence, rising from around 1% prevalence on 1 March to over 70% in the beginning of May. The researchers from the Catholic University of Leuven in Belgium found 50% of infections due to B.1.617.2 variant in England as on 24 May 2021 [55]. An analysis of UK sequencing data suggests that numbers of B.1.617.2 infections could be growing 13% faster than B.1.1.7 infections each day in UK.

The major variations for B.1.617.1 include G142D, E154K, L452R, E484Q, D614G, P681R and Q1071H. While T19R, L452R, E484Q, D614G, P681R and D950N variations were suggested in B.1.617.3 variant. B.1.617.2, a highly infectious strain amongst these three, has shown T19R, G142D, del157/158, L452R, T478K, D614G, P681R and D950N mutations.

Although several countries initiated vaccination drives to fight against infections due to B.1.617, the cases still rise exponentially from mid-May 2021 to February 2022. The mutations observed in B.1.617 and sub-lineages reported potential reduced antibody efficacy and neutralization by vaccines used worldwide [56–58]. However, the cases were gradually decreased after May 2021. This might be due to an aggressive vaccination drive handled worldwide. In the study conducted by Emma et al. on 159 participants, two doses of BNT162b2 vaccine elicited anti-Wild-type spike antibodies in all participants, and NAb activity against all strains, including B.1.1.7, B.1.617.2 and B.1.351 [59]. The investigation on 28 vaccinated people with BBV152 (Covaxin) also reported the neutralizing efficiencies against B.1.617 in their serum after vaccination [60]. However, the present study provides little insight into the variations observed in B.1.617 lineage and subsequent lineages. It was explained that B.1.617 should be sub-grouped to another variation. Although B.1.617 was first reported in India, other countries reported the same mutations due to travelling and migration of peoples. However, in some countries, varied mutations were reported with diverse similarities compared to those initially reported in India. The present study provides support for clustering and subgrouping of B.1.617 strain of SARS-CoV-2.

## 5. Conclusions

The present work explains the strategic distribution of the B.1.617 lineage of (SARS-CoV-2), also known as Double Mutant Strain of SARS-CoV-2. The workflow used in the present study provides detailed information based on effects caused due to mutations from whole-genome sequences. The analysis mentioned multiple groups based on the variant analysis having the non-synonymous type of mutations observed in all genes. The results indicate a mutation observed from only Switzerland representing a unique cluster. Whereas mutations observed in Barharin, New Zealand and Australia represent the separate cluster. The results describe that B.1.617 was not spread through India to other countries but eventually observed as a sub-lineage of B.1.617.1, B.1.617.2 and B.1.617.3. The variations E154K, E484Q, L452R, P681R and Q1071H were observed in most samples with allele frequency beyond 0.85. These variations might be responsible for several cases during the wave of COVID-19 infections. The recent submissions to NCBI GenBank database and GISAID EpiFlu Database will elucidate more variations belonging to B.1.617 and its sub-lineages. The resulting continuous tracking of such variations will generate a complete picture of epidemiology and transmission of SARS-CoV-2 during the second wave of COVID-19 worldwide.

**Supplementary Materials:** The following supporting information can be downloaded at: https://www.mdpi.com/article/10.3390/covid2050038/s1, Table S1: Mutations observed in sequences identified from GISAID and Nextstrain-ncov databases.

**Author Contributions:** Conceptualization, V.M. and R.P.; methodology, V.M.; software, V.M., R.P., H.G. and U.B.; validation, V.M., P.D. and R.P.; formal analysis, V.M., P.D., A.G., N.V. and M.G.; investigation, V.M. and R.P.; resources, H.G. and U.B.; data curation, V.M., H.G., U.B. and R.P.; writing—original draft preparation, V.M. and R.P.; writing—review and editing, H.G. and R.P.; visualization, supervision. All authors have read and agreed to the published version of the manuscript.

**Funding:** This research was not received any funding support from the agency.

**Informed Consent Statement:** Not Applicable.

**Data Availability Statement:** All the data regarding the study are available at http://covid19.vnsguhpc.co.in:808 (accessed on 30 June 2021).

**Acknowledgments:** We have to express our appreciation to the GISAID and submitter for providing us whole-genome sequences. We are also immensely grateful to the Supercomputing facility, Veer Narmad South Gujarat University, to analyze our data in the available supercomputing facility.

**Conflicts of Interest:** Authors declare no conflict of interest in the present study.

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
