# Peer review of "Variant Analysis and Strategic Clustering to Sub-Lineage of Double Mutant Strain B.1.617 of SARS-CoV-2"

_covid, doi:10.3390/covid2050038_

Round 1

Reviewer 1 Report

The authors investigated 822 genome sequences developed by GISAID last year and reported the viral evolution and genome variations of B.1.617 lineage. Which will be useful toward antibody/antivirals development and played an important role in the SARS-CoV-2 migration and outbreak response improvement. The detailed comments can be seen in attached report.

Reviewer 2 Report

This study's finding represented that “double mutant” strain is not only spread through traveling but it is also observed to evolve naturally with different mutations observed in B.1.617 lineage. The information extracted from the study helps to understand viral evolution and genome variations of B.1.617 lineage. The results support the need of separating B.1.617 into sub-lineages.

The manuscript is well written and topic is very interesting, I suggest to publish without change.

The authors also can consider this ref PMID: 33134310 in the introduction part

Reviewer 3 Report

The authors conducted analysis of 822 genome sequences of double mutant strain (B.1.617) worldwide. They have compared the SARS-CoV-2 genomes of B.1617 lineage to reference Wuhan’s seafood market genome sequence with the aim of gain important insight into virus mutations. They analyzed a bunch of data of B.1617, and the resulting information might be important for understanding viral evolution. However, it is difficult to get a story of this article. The authors may reconstruct their manuscript by clarifying their purpose of the study and new points of their data. Unfortunately, I could not get reason why they analyzed the data and what is informative. I felt an impression that the authors just put what they did thorough their work.

Based on the above, I showed my comments as bellow.

Major comments

  • Authors need to clarify their purpose and new points from the data. Please make sure it and what information is extracted from their analysis.
  • Authors need select their data based on new findings from their analysis. Otherwise, the current manuscript is just presenting their data.
  • In Introduction, the authors may need a paragraph explaining genome structure of SARS-CoV-2.
  • The authors need to put appropriate captions in all figures. There are no explanations, therefore, it is tough to understand how to see them.
  • Especially, the author should explain Figs 3-9 more. I could not understand how to see them and what do they mean. They are just put on manuscript and make their points complex.
  • Especially in Discussion, the manuscript likes as a review paper and redundant. Please clarify new findings of their data and interesting points by comparing past studies.

Minor point

  • “ClustalO” means “Clustal omega”?

Round 2

Reviewer 3 Report

It is improved.